# Discovering Reinforcement Learning Algorithms

Junhyuk Oh[*]        Matteo Hessel        Wojciech M. Czarnecki        Zhongwen Xu

Hado van Hasselt        Satinder Singh        David Silver

DeepMind

## Abstract

Reinforcement learning (RL) algorithms update an agent's parameters according to one of several possible rules, discovered manually through years of research. Automating the discovery of update rules from data could lead to more efficient algorithms, or algorithms that are better adapted to specific environments. Although there have been prior attempts at addressing this significant scientific challenge, it remains an open question whether it is feasible to discover alternatives to fundamental concepts of RL such as value functions and temporal-difference learning. This paper introduces a new meta-learning approach that discovers an entire update rule which includes both 'what to predict' (e.g. value functions) and 'how to learn from it' (e.g. bootstrapping) by interacting with a set of environments. The output of this method is an RL algorithm that we call Learned Policy Gradient (LPG). Empirical results show that our method discovers its own alternative to the concept of value functions. Furthermore it discovers a bootstrapping mechanism to maintain and use its predictions. Surprisingly, when trained solely on toy environments, LPG generalises effectively to complex Atari games and achieves non-trivial performance. This shows the potential to discover general RL algorithms from data.

## 1   Introduction

Reinforcement learning (RL) has a clear objective: to maximise expected cumulative rewards (or average rewards), which is simple, yet general enough to capture many aspects of intelligence. Even though the objective of RL is simple, developing efficient algorithms to optimise such objective typically involves a tremendous research effort, from building theories to empirical investigations. An appealing alternative approach is to automatically discover RL algorithms from data generated by interaction with a set of environments, which can be formulated as a meta-learning problem. Recent work has shown that it is possible to meta-learn a policy update rule when given a value function, and that the resulting update rule can generalise to similar or unseen tasks (see Table 1).

However, it remains an open question whether it is feasible to discover fundamental concepts of RL entirely from scratch. In particular, a defining aspect of RL algorithms is their ability to learn and utilise value functions. Discovering concepts such as value functions requires an understanding of both 'what to predict' and 'how to make use of the prediction'. This is particularly challenging to discover from data because predictions only have an indirect effect on the policy over the course of multiple updates. We hypothesise that a method capable of discovering value functions for itself may also discover other useful concepts, potentially opening up entirely new approaches to RL.

Motivated by the aforementioned open questions, this paper takes a step towards discovering general-purpose RL algorithms. We introduce a meta-learning framework that jointly discovers both 'what the agent should predict' and 'how to use predictions for policy improvement' from data generated by interacting with a distribution of environments. Our architecture, Learned Policy Gradient (LPG), does not enforce any semantics on the agent's vector-valued outputs but instead allows the update

---

[*]Corresponding author: junhyuk@google.com

Table 1: Methods for discovering RL algorithms.

| Algorithm | Discovery | Method | Generality | Train | Test |
|---|---|---|---|---|---|
| RL$^2$ [9, 38] | N/A | $\nabla$ | Domain-specific | 3D maze | Similar 3D maze |
| EPG [18] | $\hat{\pi}$ | ES | Domain-specific | MuJoCo | Similar MuJoCo |
| ML$^3$ [6] | $\hat{\pi}$ | $\nabla\nabla$ | Domain-specific | MuJoCo | Similar MuJoCo |
| MetaGenRL [19] | $\hat{\pi}$ | $\nabla\nabla$ | General | MuJoCo | Unseen MuJoCo |
| **LPG** | $\hat{\pi}, \hat{y}$ | $\nabla\nabla$ | General | Toy | Unseen Atari |

$\hat{\pi}$: policy update rule, $\hat{y}$: prediction update rule (i.e., semantics of agent's prediction).
$\nabla$: gradient descent, $\nabla\nabla$: meta-gradient descent, ES: evolutionary strategy.

rule (i.e., the meta-learner) to decide what this vector should be predicting. We then propose a meta-learning framework to discover such update rule from multiple learning agents, each of which interacts with a different environment.

Experimental results show that our algorithm can discover useful functions, and use those functions effectively to update the agents policy. Furthermore, empirical analysis shows that the discovered functions converge towards an encoding of a notion of value function, and furthermore maintain this value function via a form of bootstrapping. We also evaluated the ability of the discovered RL algorithm to generalise to new environments. Surprisingly, even though the update rule was discovered solely from interactions with a very small set of toy environments, it was able to generalise to a number of complex Atari games [3], as shown in Figure 9. To our knowledge, this is the first to show that it is possible to discover an entire update rule, and that the update rule discovered from toy domains can be competitive with human-designed algorithms on a challenging benchmark.

## 2 Related Work

**Early Work on Learning to Learn** The idea of learning to learn has been discussed for a long time with various formulations such as improving genetic programming [30], learning a neural network update rule [4], learning rate adaptations [33], self-weight-modifying RNNs [31], and transfer of domain-invariant knowledge [35]. Such work showed that it is possible to learn not only to optimise fixed objectives, but also to improve the way to optimise at a meta-level.

**Learning to Learn for Few-Shot Task Adaptation** Learning to learn has received much attention in the context of few-shot learning [29, 37]. MAML [11, 12] allows to meta-learn initial parameters by backpropagating through the parameter updates. RL$^2$ [9, 38] formulates learning itself as an RL problem by unrolling LSTMs [17] across the agent's entire lifetime. Other approaches include simple approximation [27], RNNs with Hebbian learning [23, 24], and gradient preconditioning [13]. All these do not clearly separate between agent and algorithm, thus, the resulting meta-learned algorithms are specific to a single agent architecture by definition of the problem.

**Learning to Learn for Single Task Online Adaptation** A different corpus of work focuses on learning to learn a single task within a single lifetime. Xu et al. [41] introduced the meta-gradient RL approach; this uses backpropagation through the agent's updates, to calculate the gradient with respect to the meta-parameters of the update. This approach has been applied to meta-learn various forms of algorithmic components such as the discount factor [41], intrinsic rewards [44], auxiliary tasks [36], returns [39], auxiliary policy updates [45], off-policy corrections [42], and update target [40]. In contrast, our work has an orthogonal goal: to discover general algorithms that are effective for a broader class of agents and environments instead of being adaptive to a particular environment.

**Discovering Reinforcement Learning Algorithms** There have been a few attempts to meta-learn RL algorithms, from earlier work on bandit algorithms [22, 21] to curiosity algorithms [1] and RL objectives [18, 43, 6, 19] (see Table 1 for comparison). EPG [18] uses an evolutionary strategy to find a policy update rule. Zheng et al. [43] showed that general knowledge for exploration can be meta-learned in the form of reward function. ML$^3$ [6] meta-learns a loss function using meta-gradients. However, the prior work can generalise only up to similar tasks within the same domain. Most recently, MetaGenRL [19] was proposed to meta-learn a domain-invariant policy update rule, capable of generalising from a few MuJoCo environments to other MuJoCo environments. However, no prior work has attempted to discover the *full* update rule; instead they all relied on value functions, arguably

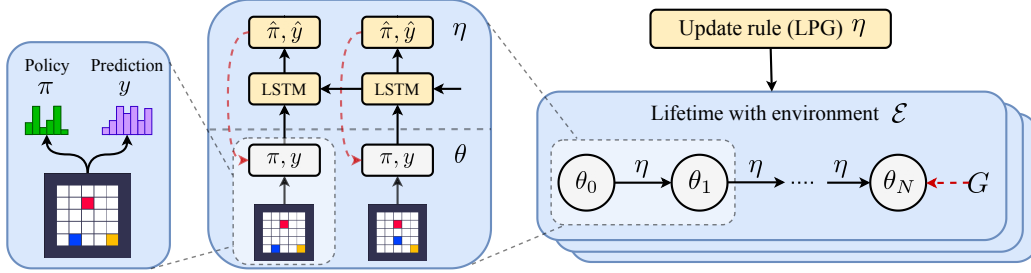

Figure 1: Meta-training of learned policy gradient (LPG). (Left) The agent parameterised by $\theta$ produces action-probabilities $\pi$ and a prediction vector $y$ for a state. (Middle) The update rule (LPG) parameterised by $\eta$ takes the agent outputs as input and unrolls an LSTM backward to produce targets for the agent outputs ($\hat{\pi}, \hat{y}$). (Right) The update rule $\eta$ is meta-learned from multiple lifetimes, in each of which a distinct agent interacts with an environment sampled from a distribution, and updates its parameters $\theta$ using the shared update rule. The meta-gradient is calculated to maximise the return after every $K < N$ parameter updates by sliding window, averaged over all parallel lifetimes.

the most fundamental building block of RL, for bootstrapping. In contrast, our LPG meta-learns its own mechanism for bootstrapping. Additionally, this paper is the first to show that a radical generalisation from toy environments to a challenging benchmark is possible.

The goal of this work is similar to the second pillar (meta-learning learning algorithms) in AI-GAs [7]. However, we aim to achieve generalisation not just across tasks but also across different domains. Learning domain-invariant algorithms has been discussed in the context of discovering an internal structure of agent (e.g., neural update rule [28], RNN update rule [14]). In contrast, the discovered update rule in our work is independent of agent's parameterisation.

## 3   Meta-Learning Framework for Learned Policy Gradient

The goal of the proposed meta-learning framework is to find the optimal update rule, parameterised by $\eta$, from a distribution of environments $p(\mathcal{E})$ and initial agent parameters $p(\theta_0)$:

$$\eta^* = \arg\max_{\eta} \mathbb{E}_{\mathcal{E} \sim p(\mathcal{E})} \mathbb{E}_{\theta_0 \sim p(\theta_0)} [G], \tag{1}$$

where $G = \mathbb{E}_{\pi_{\theta_N}}[\sum_t^\infty \gamma^t r_t]$ is the expected return at the end of the lifetime. Intuitively, the objective aims to find an update rule $\eta$ such that when it is used to update the agent's parameters until the end of its lifetime ($\theta_0 \to \cdots \to \theta_N$), the agent maximises the expected return in the given environment. The resulting update rule is called Learned Policy Gradient (LPG). The overview of meta-training process is summarised in Figure 1 and Algorithm 1.

### 3.1   LPG Architecture

As illustrated in Figure 1, LPG is an update rule parameterised by meta-parameters $\eta$ which requires an agent to produce a policy $\pi_\theta(a|s)$ and a $m$-dimensional categorical prediction vector $y_\theta(s) \in [0,1]^m$. The LPG is a backward LSTM [17] network that produces as output how to update the policy and the prediction vector $\hat{\pi} \in \mathbb{R}, \hat{y} \in [0,1]^m$ from the trajectory of the agent. More specifically, it takes $x_t = [r_t, d_t, \gamma, \pi_\theta(a_t|s_t), y_\theta(s_t), y_\theta(s_{t+1})]$ at each time-step $t$, where $r_t$ is a reward, $d_t$ is a binary value indicating episode-termination, and $\gamma$ is a discount factor. By construction, LPG is invariant to observation space and action space, as it does not take them as input. Instead, it only takes the probability of the chosen action $\pi(a|s)$. This structure allows the LPG architecture to be applicable to entirely different environments while preventing overfitting.

### 3.2   Agent Update ($\theta$)

Agent parameters are updated by performing gradient ascent in the direction of:

$$\Delta\theta \propto \mathbb{E}_{\pi_\theta} [\nabla_\theta \log \pi_\theta(a|s)\hat{\pi} - \alpha_y \nabla_\theta D_{\mathrm{KL}}(y_\theta(s)\|\hat{y})], \tag{2}$$

where $\hat{\pi}$ and $\hat{y}$ are the output of LPG. $D_{\mathrm{KL}}(P\|Q) = \sum_x P(x) \log \frac{P(x)}{Q(x)}$ is the Kullback–Leibler divergence. $\alpha_y$ is a coefficient for the prediction update respectively. At a high-level, $\hat{\pi}$ specifies how the action-probability should be adjusted, and has a direct effect on the agent's behaviour. $\hat{y}$ specifies

---

**Algorithm 1** Meta-Training of Learned Policy Gradient

---

**Input**: $p(\mathcal{E})$: Environment distribution, $p(\theta_0)$: Initial agent parameter distribution
Initialise meta-parameters $\eta$ and hyperparameter sampling distribution $p(\alpha|\mathcal{E})$
Sample batch of environment-agent-hyperparameters $\{\mathcal{E} \sim p(\mathcal{E}), \theta \sim p(\theta_0), \alpha \sim p(\alpha|\mathcal{E})\}_i$
**repeat**
    **for all** lifetimes $\{\mathcal{E}, \theta, \alpha\}_i$ **do**
        Update parameters $\theta$ using $\eta$ and $\alpha$ for $K$ times using Eq. (2)
        Compute meta-gradient using Eq. (4)
        **if** lifetime ended **then**
            Update hyperparameter sampling distribution $p(\alpha|\mathcal{E})$
            Reset lifetime $\mathcal{E} \sim p(\mathcal{E}), \theta \sim p(\theta_0), \alpha \sim p(\alpha|\mathcal{E})$
        **end if**
    **end for**
    Update meta-parameters $\eta$ using the meta-gradients averaged over all lifetimes.
**until** $\eta$ converges

---

a target categorical distribution that the agent should predict for a given state, and does not have an effect on the policy until the LPG discovers useful semantics (e.g., value function) of it and uses $y$ to indirectly change the policy by bootstrapping, which makes the discovery problem challenging.

Note that the proposed framework is not restricted to this particular form of agent update and architecture (e.g., categorical prediction with KL-divergence). We explore this specific form partly inspired by the success of Distributional RL [2, 8]. However, we do not enforce any semantics on $y$ but allow the LPG to discover the semantics of $y$ from data.

### 3.3  LPG Update ($\eta$)

LPG is meta-trained by taking into account how much it improves the performances of a population of agents interacting with different kinds of environments. Specifically, the meta-gradients are calculated by applying policy gradient to the objective in Eq. (1) as follows:

$$\Delta\eta \propto \mathbb{E}_{\mathcal{E}}\mathbb{E}_{\theta_0}\left[\nabla_\eta \log \pi_{\theta_N}(a|s)G\right] \tag{3}$$

Intuitively, we perform parameter updates for $N$ times using the update rule $\eta$ from $\theta_0$ to $\theta_N$ until the end of the lifetime and estimate policy gradient for the updated parameters $\theta_N$ to find the meta-gradient direction that maximises the expected return ($G$) of $\theta_N$. This requires backpropagation through the agent's update process as in [41, 12]. In practice, due to memory constraints, we consider a smaller sliding window, and perform a truncated backpropagation every $K < N$ parameter updates.

**Regularisation**  We find that the optimisation can be very hard and unstable, mainly because the LPG needs to learn an appropriate semantics of predictions $\hat{y}$, as well as learning to use predictions $y$ properly for bootstrapping without access to the value function. To stabilise training, we propose to add the following regularisers (on the targets $\hat{\pi}$ and $\hat{y}$), resulting in the meta-gradient:

$$\mathbb{E}_{\mathcal{E}}\mathbb{E}_{\theta_0}\left[\nabla_\eta \log \pi_{\theta_N}(a|s)G + \beta_0\nabla_\eta\mathcal{H}(\pi_{\theta_N}) + \beta_1\nabla_\eta\mathcal{H}(y_{\theta_N}) - \beta_2\nabla_\eta\|\hat{\pi}\|_2^2 - \beta_3\nabla_\eta\|\hat{y}\|_2^2\right], \tag{4}$$

where $\mathcal{H}(\cdot)$ is the entropy, and $\{\beta_i\}$ are meta-hyperparameters for each regularisation term. $\mathcal{H}(y)$ penalises too deterministic predictions, which shares the same motivation with policy entropy regularisation $\mathcal{H}(\pi)$ [25]. These are not applied to the agent but applied to the update rule so that the resulting LPG has such properties. The L2-regularisation for $\hat{\pi}, \hat{y}$ prevents the updates from being too aggressive. We discuss the effect of these regularisers in Section 4.4.

### 3.4  Balancing Agent Hyperparameters for Stabilisation ($\alpha$)

While previous approaches [6, 19] used fixed agent hyperparameters (e.g., learning rate) during meta-training, we find it problematic when meta-training across entirely different environments. For example, if the learning rate used for environment A happens to be relatively larger than that for environment B, the optimal scale of $\hat{\pi}$ should be smaller for A and larger for B. Since the update rule $\eta$ is environment-agnostic, it would get contradicting meta-gradients between two environments, making meta-training unstable. Furthermore, it is impossible to pre-balance hyperparameters due to their dependence on $\eta$, which changes during meta-training making the problem of balancing

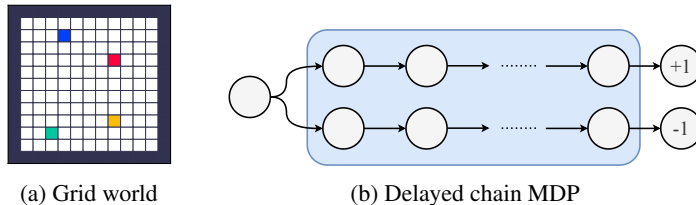

|(a) Grid world|(b) Delayed chain MDP|

Figure 2: Training environments. (a) The agent receives the corresponding rewards by collecting objects. (b) The first action determines the reward at the end of the episode.

hyperparameters inherently non-stationary. To address this, we modify the objective in Eq. 1 to:

$$\eta^* = \arg\max_\eta \mathbb{E}_{\mathcal{E}\sim p(\mathcal{E})} \max_\alpha \mathbb{E}_{\theta_0\sim p(\Theta)} [G], \tag{5}$$

where $\alpha = \{\alpha_{\text{lr}}, \alpha_y\}$ are a learning rate and a coefficient for prediction update (see Eq. (2)). This objective seeks the optimal update rule given the optimal hyperparameters for each environment. To optimise this, in practice, we propose to use a bandit $p(\alpha|\mathcal{E})$ that samples hyperparameters for each lifetime and updates the sampling distribution according to the return at the end of each lifetime. By making $p(\alpha|\mathcal{E})$ adapt to each environment, hyperparameters are automatically balanced across environments, which makes meta-gradient less noisy. Note that this is done only during meta-training. During meta-testing on unseen environments, hyperparameters need to be manually selected in the same way that we do for existing RL algorithms. Further details on how this was done in our experiments are described in the supplementary material.

## 4 Experiment

The experiments are designed to answer the following research questions:

- Can LPG discover a useful semantics of predictions for efficient bootstrapping?
- What are the discovered semantics of predictions?
- How crucial is it to discover the semantics of predictions?
- How crucial are the regularisers and hyperparameter balancing?
- Can LPG generalise from toy environments to complex Atari games?

### 4.1 Experimental Setup

**Training Environments**    For meta-training of LPG, we introduce three different kinds of toy domains as illustrated Figure 2. *Tabular grid worlds* are grid worlds with fixed object locations. *Random grid worlds* have randomised object locations for each episode. *Delayed chain MDPs* are simple MDPs with delayed rewards. There are 5 variations of environments for each domain with various number of rewarding states and episode lengths. The training environments are designed to captures basic RL challenges such as delayed reward, noisy reward, and long-term credit assignment. Most of the training environments are tabular without involving any function approximators. The details of all environments are described in the supplementary material.

**Implementation Details**    We used a 30-dimensional prediction vector $y \in [0, 1]^{30}$. During meta-training, we updated the agent parameters after every 20 time-steps. Since most of the training episodes span over 20-2000 steps, LPG must discover a long-term semantics for the predictions $y$ to be able to maximise long-term future rewards from partial trajectories. The algorithm is implemented using JAX [5]. More implementation details are described in the supplementary material.

**Baselines**    As discussed in Section 2 and Table 1, most of the prior work does not support generalisation across entirely different environments except MetaGenRL [19]. However, MetaGenRL is designed for continuous control and based on DDPG [32, 20]. Instead, to investigate the importance of discovering prediction semantics, we compare to our own baseline **LPG-V**, a variant of LPG that, like MetaGenRL, only learns a policy update rule ($\hat{\pi}$) given a value function trained by TD($\lambda$) [34] without discovering its own prediction semantics.[2] Additionally, we also compare against an advantage actor-critic (A2C) [25] as a canonical human-discovered algorithm baseline.

[2]LPG-V does not fully represent MetaGenRL as it includes other advances introduced in this paper.

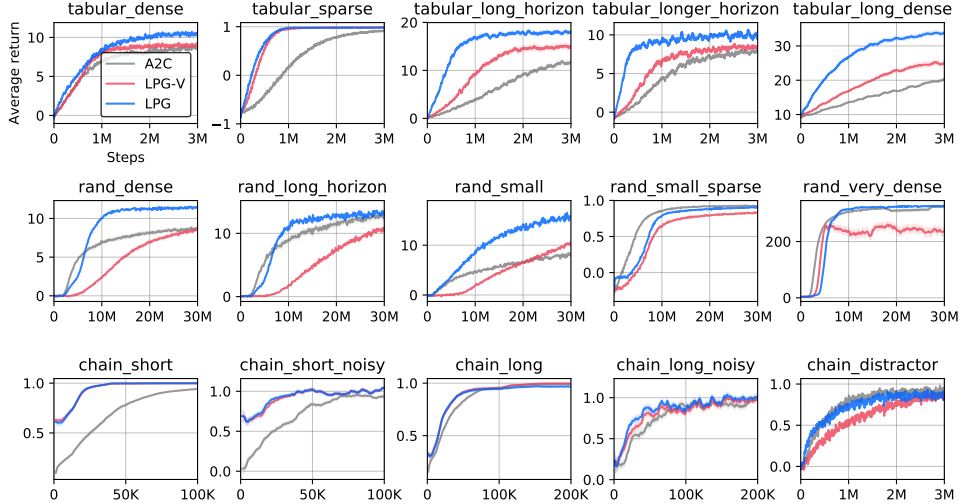

Figure 3: Evaluation on the training environments. Shaded areas show standard errors from 64 random seeds.

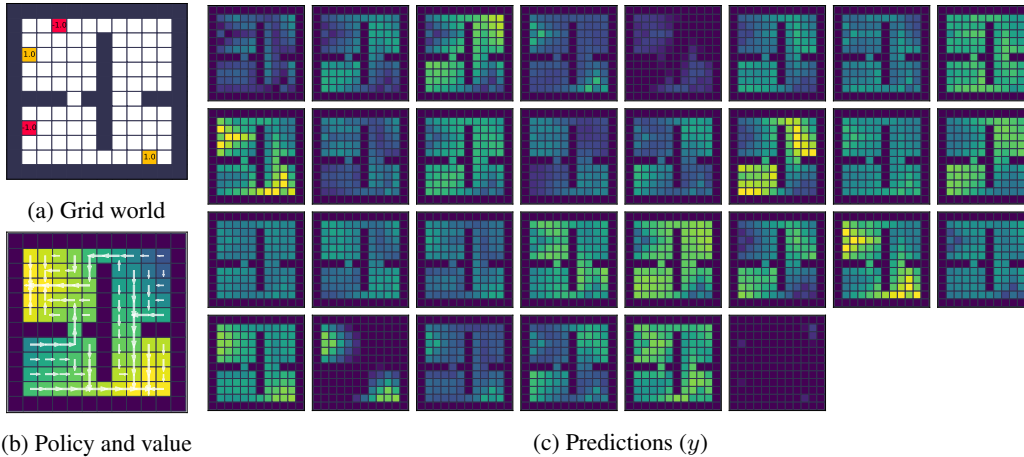

(a) Grid world

(b) Policy and value

(c) Predictions ($y$)

Figure 4: Visualisation of predictions. (a) A grid world with positive goals (yellow) and negative goals (red). (b) A near-optimal policy and its true values. (c) Visualisation of $y \in [0, 1]^{30}$ for the given policy in (b).

## 4.2 Specialising in Training Environments

We evaluated the LPG on the training environments to see whether LPG has discovered an effective update rule. The result in Figure 3 shows that the LPG outperforms A2C on most of the training environments. This shows that the proposed framework can discover an update rule that outperforms the *outer* algorithm used for discovery (i.e., policy gradient in Eq (4)) on the given environments. In addition, the result suggests that LPG specialises in certain classes of environments, and that LPG can be potentially an even better solution than hand-designed RL algorithms if one is interested in a specific class of RL problems. On the other hand, LPG-V is much worse than LPG while not clearly better than A2C. This shows that discovering the semantics of prediction is the key for the performance, which justifies our approach in contrast to the prior work that only learns a policy update rule while relying on grounded value functions (see Table 1 for comparison).

## 4.3 Analysis of Learned Policy Gradient

**What does the prediction ($y$) look like?**    Since the discovered semantics of prediction is the key for the performance, a natural question is what are the discovered concepts and how they work. To answer this, we first visualised the predictions for a given tabular grid world instance and for a fixed policy in Figure 4. Specifically, we updated only $y$ using the LPG while fixing the policy parameters,

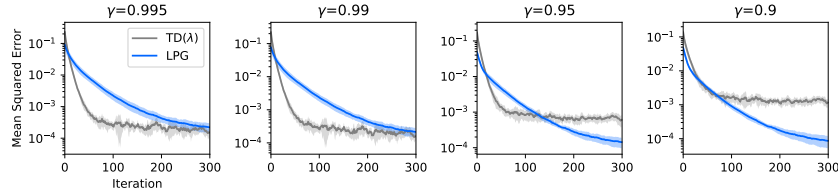

Figure 5: Value regression from predictions over the course of policy evaluation. Each plot shows mean squared errors to true values at various discount factors averaged over held-out 10 grid world instances.

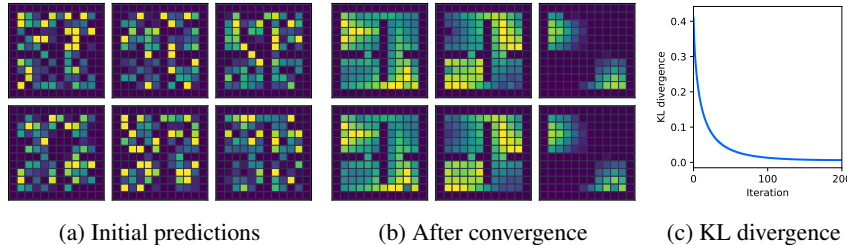

(a) Initial predictions      (b) After convergence      (c) KL divergence

Figure 6: Convergence of predictions. (a-b) An example of two randomly initialised predictions $(y_0, y_1)$ are shown at the top $(y_0)$ and the bottom $(y_1)$ (a) before and (b) after convergence. 3-dimensions are selected for visualisation. (c) The progression of $D_{\text{KL}}(y_0||y_1)$ averaged over 128 grid world instances.

which is analogous to policy evaluation.[3] The visualisation in Figure 4c shows that some predictions have large values around positive rewarding states, and they are propagated to nearby states similarly to the true values in Figure 4b. This visualisation implicitly shows that the LPG is asking the agent to predict future rewards and use such information for bootstrapping.

**Does the prediction ($y$) capture true values and beyond?** To further investigate how rich the predictions are, we generated $y$ vectors as in Figure 4c for many grid world instances with a discount factor of 0.995 and trained a value regression model $g : \mathbb{R}^{30} \mapsto \mathbb{R}$, a 1-layer multi-layer perceptron (MLP), to predict true values just from predictions $y$ for various discount factors from 0.995 to 0.9. We then evaluated how accurate the value regression is on a held-out set of grid worlds. For a comparison, we also trained a value regression model $h : \mathbb{R} \mapsto \mathbb{R}$ for TD($\lambda$) from values at discount factor 0.995 to values at the other discount factors.

Interestingly, the result in Figure 5 shows that the value regression from $y$ is almost as good as TD($\lambda$) at the original discount factor (0.995) as updated by the LPG, which implies that the information in $y$ is rich enough to recover the original concept of value function. More interestingly, Figure 5 also shows that $y$ captures true values at lower discount factors, even though it was generated with a discount factor of 0.995. On the other hand, the information in the scalar value with TD($\lambda$) is too limited to capture values at lower discount factors. This result suggests that the proposed framework can automatically discover a rich and useful semantics of predictions that can almost recover the value functions at various horizons, even though such a semantics was not enforced during meta-training.

**Does the prediction ($y$) converge?** The convergence of the predictions learned by different RL methods is one of their most critical properties. Classical algorithms, such as temporal-difference (TD) learning, have a convergence guarantee to a precisely defined semantics (i.e., the expected return) in tabular settings [34]. On the other hand, LPG does not have such a guarantee, because the prediction semantics is meta-learned with the sole aim to improve the performance of the agent, which means that LPG can, in principle, contain non-convergent dynamical systems that could cycle or diverge. Therefore, we empirically investigate the convergence property of LPG. The result in Figure 6 shows that two different prediction vectors $(y_0, y_1)$ converge to almost the same values when updated by the LPG. This implies that a stationary semantics, to which predictions converge, has naturally emerged from the proposed framework even without any theoretical constraint.

## 4.4 Ablation Study

As discussed in Section 3.2, we find that meta-training is very hard and unstable as it is required to discover the entire update rule. Figure 7 summarises the effect of each idea introduced by this paper.

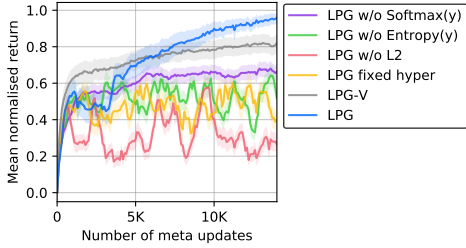

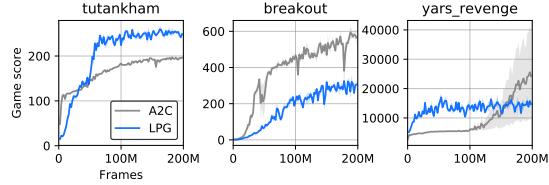

Figure 7: Ablation study. Each curve shows normalised return ($G_{\text{norm}} \in [0, 1]$) averaged over 15 training environments throughout meta-training.

Figure 8: Example learning curves on Atari games. LPG outperforms A2C on `tutankham`, learns slowly on `breakout`, and prematurely converges to a sub-optimal policy on `yars-revenge`. The learning curves across all Atari games are available in the supplementary material.

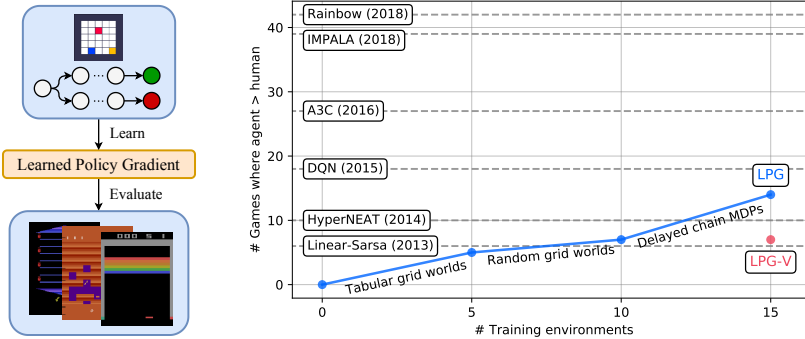

Figure 9: Generalisation from toy environments to Atari. X-axis is the number of toy environments used to meta-learn the LPG. Y-axis is the number of Atari games where the agent outperforms humans at the end of training. Dashed lines correspond to state-of-the-art algorithms for each year [3, 15, 26, 25, 10, 16].[4]

'LPG w/o Softmax(y)' uses $y \in \mathbb{R}^{30}$ without softmax but with $\|y - \hat{y}\|_2^2$ instead of KL-divergence in Eq. (2). 'LPG w/o Entropy(y)' and 'LPG w/o L2' are without entropy regularisation of $y$ and without L2 regularisation of $\hat{\pi}, \hat{y}$ in Eq. (4) respectively. 'LPG fixed hyper' is trained with fixed hyperparameters for each training environment instead of balancing them during meta-training as introduced in Section 3.4. The result in Figure 7 shows that all of these ideas are crucial for the performance, and training tends to be very unstable without either of them. On the other hand, we found that meta-training of LPG-V is stable even without regularisers. However, LPG-V converges to a sub-optimal update rule, whereas LPG eventually finds a better update rule by discovering what to predict. This result supports our hypothesis that discovering alternatives to value functions has the greater potential to find better update rules, although optimisation can be more difficult.

### 4.5 Generalising from Toy Environments to Atari Games

To see how general LPG can be when discovered solely from toy environments, we evaluated the LPG directly on complex Atari games. As summarised in Figure 9, the LPG generalises to Atari games reasonably well when compared to the advanced RL algorithms. This is surprising in that the training environments consist of mostly tabular environments with basic tasks that are much simpler than Atari games, and the LPG has never seen such complex domains during meta-training. Nevertheless, the agents trained with the LPG can learn complex behaviours across many Atari games achieving super-human performances on 14 games, without relying on any hand-designed RL components such as value function but rather using its own update rule discovered from scratch.

We found that specific types of training environments, such as delayed chain MDPs, significantly improved the generalisation performance (see Figure 9). This suggests that there may be a small but carefully designed set of environments that capture important challenges in RL so that when they are used for meta-training, the resulting LPG is general enough to perform well across many complex domains.

Although LPG is still behind the advanced RL algorithms such as A2C, the fact that LPG outperforms A2C on not just the training environments but also a few Atari games (see Figure 8 for example) implies that LPG specialises in a particular type of RL problems instead of being strictly worse than A2C. On the other hand, Figure 9 shows that the generalisation performance improves quickly as

the number of training environments grows, which suggests that it may be feasible to discover a general-purpose RL algorithm once a larger set of environments are available for meta-training.

## 5 Conclusion

This paper made the first attempt to meta-learn a full RL update rule by jointly discovering both 'what to predict' and 'how to bootstrap', replacing existing RL concepts such as value function and TD-learning. The results from a small set of toy environments showed that the discovered LPG maintains rich information in the prediction, which was crucial for efficient bootstrapping. We believe this is just the beginning of the fully data-driven discovery of RL algorithms; there are many promising directions to extend our work, from procedural generation of environments, to new advanced architectures and alternative ways to generate experience. The radical generalisation from the toy domains to Atari games shows that it may be feasible to discover an efficient RL algorithm from interactions with environments, which would potentially lead to entirely new approaches to RL.

## Broader Impact

The proposed approach has a potential to dramatically accelerate the process of discovering new reinforcement learning (RL) algorithms by automating the process of discovery in a data-driven way. If the proposed research direction succeeds, this could shift the research paradigm from manually developing RL algorithms to building a proper set of environments so that the resulting algorithm is efficient.

Additionally, the proposed approach may also serve as a tool to assist RL researchers in developing and improving their hand-designed algorithms. In this case, the proposed approach can be used to provide insights about what a good update rule looks like depending on the architecture that researchers provide as input, which could speed up the manual discovery of RL algorithms.

On the other hand, due to the data-driven nature of the proposed approach, the resulting algorithm may capture unintended bias in the training set of environments. In our work, we do not provide domain-specific information except rewards when discovering an algorithm, which makes it hard for the algorithm to capture bias in training environments. However, more work is needed to remove bias in the discovered algorithm to prevent potential negative outcomes.

## Acknowledgement

We thank Simon Osindero and Doina Precup for their helpful feedback on the manuscript.

## Footnotes

[3]To avoid overfitting, we created an unseen grid world task with an unseen action space.

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
