[Supplementary Material]

# Supplementary Material:
# Discovering Reinforcement Learning Algorithms

Junhyuk Oh          Matteo Hessel          Wojciech M. Czarnecki          Zhongwen Xu

Hado van Hasselt          Satinder Singh          David Silver

DeepMind

## A    Training Environments

### A.1    Tabular Grid World

When an agent collects an object, it receives the corresponding reward $r$, and the episode terminates with a probability of $\epsilon_{\text{term}}$ associated with the object. The object disappears when collected, and reappears with a probability of $\epsilon_{\text{respawn}}$ for each time-step. In the following sections, we describe each object type $i$ as $\{N \times [r, \epsilon_{\text{term}}, \epsilon_{\text{respawn}}]\}_i$, where $N$ is the number of objects with type $i$.

**Observation Space**    In tabular grid worlds, object locations are randomised across lifetimes but fixed within a lifetime. Thus, there are only $p \times 2^{\text{m}}$ possible states in each lifetime, where $p$ is the number of possible positions, and $m$ is the total number of objects. An agent is simply represented by a table with distinct $\pi(a|s)$ and $y(s)$ values for each state without any function approximation.

**Action Space**    There are two different action spaces. One version consists of 9 movement actions for adjacent positions (including staying at the same position) and 9 actions for collecting objects at adjacent positions. The other version has only 9 movement actions. In this version, an object is automatically collected when the agent visits it. We randomly sample either one of the action spaces for each lifetime during meta-training.

### A.1.1    Dense

| Component | Description |
|---|---|
| Observation | State index (integer) |
| Number of actions | 9 or 18 |
| Size | $11 \times 11$ |
| Objects | $2 \times [1, 0, 0.05], [-1, 0.5, 0.1], [-1, 0, 0.5]$ |
| Maximum steps per episode | 500 |

### A.1.2  Sparse

| Component | Description |
| --- | --- |
| Observation | State index (integer) |
| Number of actions | 9 or 18 |
| Size | $13 \times 13$ |
| Objects | $[1, 1, 0], [-1, 1, 0]$ |
| Maximum steps per episode | 50 |

### A.1.3  Long Horizon

| Component | Description |
| --- | --- |
| Observation | State index (integer) |
| Number of actions | 9 or 18 |
| Size | $11 \times 11$ |
| Objects | $2 \times [1, 0, 0.01], 2 \times [-1, 0.5, 1]$ |
| Maximum steps per episode | 1000 |

### A.1.4  Longer Horizon

| Component | Description |
| --- | --- |
| Observation | State index (integer) |
| Number of actions | 9 or 18 |
| Size | $7 \times 9$ |
| Objects | $2 \times [1, 0.1, 0.01], 5 \times [-1, 0.8, 1]$ |
| Maximum steps per episode | 2000 |

### A.1.5  Long Dense

| Component | Description |
| --- | --- |
| Observation | State index (integer) |
| Number of actions | 9 or 18 |
| Size | $11 \times 11$ |
| Objects | $4 \times [1, 0, 0.005]$ |
| Maximum steps per episode | 2000 |

## A.2  Random Grid World

The random grid worlds are almost the same as the tabular grid worlds except that object locations are randomised within a lifetime. More specifically, object locations are randomly determined at the beginning of each episode, and objects re-appear at random locations after being collected. Due to the randomness, the state space is exponentially large, which requires function approximation to represent an agent. The observation consists of a tensor $\{0, 1\}^{N \times H \times W}$, where $N$ is the number of object types, $H \times W$ is the size of the grid.

### A.2.1 Dense

| Component | Description |
|---|---|
| Observation | $\{0,1\}^{N \times H \times W}$ |
| Number of actions | 9 or 18 |
| Size | $11 \times 11$ |
| Objects | $2 \times [1, 0, 0.05], [-1, 0.5, 0.1], [-1, 0, 0.5]$ |
| Maximum steps per episode | 500 |

### A.2.2 Long Horizon

| Component | Description |
|---|---|
| Observation | $\{0,1\}^{N \times H \times W}$ |
| Number of actions | 9 or 18 |
| Size | $11 \times 11$ |
| Objects | $2 \times [1, 0, 0.01], 2 \times [-1, 0.5, 1]$ |
| Maximum steps per episode | 1000 |

### A.2.3 Small

| Component | Description |
|---|---|
| Observation | $\{0,1\}^{N \times H \times W}$ |
| Number of actions | 9 or 18 |
| Size | $5 \times 7$ |
| Objects | $2 \times [1, 0, 0.05], 2 \times [-1, 0.5, 0.1]$ |
| Maximum steps per episode | 500 |

### A.2.4 Small Sparse

| Component | Description |
|---|---|
| Observation | $\{0,1\}^{N \times H \times W}$ |
| Number of actions | 9 or 18 |
| Size | $5 \times 7$ |
| Objects | $[1, 1, 1], 2 \times [-1, 1, 1]$ |
| Maximum steps per episode | 50 |

### A.2.5 Very Dense

| Component | Description |
|---|---|
| Observation | $\{0,1\}^{N \times H \times W}$ |
| Number of actions | 9 or 18 |
| Size | $11 \times 11$ |
| Objects | $[1, 0, 1]$ |
| Maximum steps per episode | 2000 |

## A.3 Delayed Chain MDP

This environment is inspired by the *Umbrella* environment in Behaviour Suite [7]. The agent has a binary choice $(a_0, a_1)$ for each time-step. The first action determines the reward at the end of the

episode (1 or -1). The episode terminates after a fixed number of steps (i.e., chain length), which is sampled randomly from a pre-defined range for each lifetime and fixed within a lifetime. For each episode, we randomly determine which action leads to a positive reward and sample the corresponding chain MDP. There is no state aliasing because all states are distinct. Optionally, there can be noisy rewards $\{1, -1\}$ for the states in the middle that are independent of the agent's action.

### A.3.1 Short

| Component | Description |
|---|---|
| Observation | State index (integer) |
| Number of actions | 2 |
| Chain length | $[5, 30]$ |
| Noisy rewards | No |

### A.3.2 Short and Noisy

| Component | Description |
|---|---|
| Observation | State index (integer) |
| Number of actions | 2 |
| Chain length | $[5, 30]$ |
| Noisy rewards | Yes |

### A.3.3 Long

| Component | Description |
|---|---|
| Observation | State index (integer) |
| Number of actions | 2 |
| Chain length | $[5, 50]$ |
| Noisy rewards | No |

### A.3.4 Long and Noisy

| Component | Description |
|---|---|
| Observation | State index (integer) |
| Number of actions | 2 |
| Chain length | $[5, 50]$ |
| Noisy rewards | Yes |

### A.3.5 State Distraction

In this delayed chain MDP, an observation $s_t \in \{0, 1\}^{22}$ consists of two relevant bits: whether $a_0$ is the correct action and whether the agent has chosen the correct action, and noisy bits $\{0, 1\}^{20}$ that are randomly sampled independently for all states. The agent is required to find out the relevant bits while ignoring the noisy bits in the observation.

| Component | Description |
|---|---|
| Observation | $\{0, 1\}^{22}$ |
| Number of actions | 2 |
| Chain length | $[5, 30]$ |
| Noisy rewards | No |

# B Implementation Details

## B.1 Meta-Training

We trained LPGs by simulating 960 parallel lifetimes (i.e., batch size for meta-gradients), each of which has a learning agent interacting with a sampled environment, for approximately $10^{10}$ steps of interactions in total. In each lifetime, the agent updates its parameters using a batch of trajectories generated from 64 parallel environments (i.e., batch size for agent). Each trajectory consists of 20 steps. Thus, each parameter update consists of $64 \times 20$ steps. The meta-hyperparameters used for meta-training is summarised in Table 1.

**Details of LPG Architecture**  The LPG network takes $x_t = [r_t, d_t, \gamma, \pi(a_t|s_t), y_\theta(s_t), y_\theta(s_{t+1})]$ at each time-step $t$, where $r_t$ is a reward, $d_t$ is a binary value indicating episode-termination, and $\gamma$ is a discount factor. $y_\theta(s_t)$ and $y_\theta(s_{t+1})$ are mapped to a scalar using a shared embedding network ($\varphi$): Dense(16)-Dense(1).  A backward LSTM with 256 units takes $[r_t, d_t, \gamma, \pi(a_t|s_t), \varphi(y_\theta(s_t)), \varphi(y_\theta(s_{t+1}))]$ as input and produces $\hat{\pi}$ and $\hat{y}$ as output. We slightly modified the LSTM core such that the hidden states are reset for terminal states ($d_t = 0$), which blocks information from flowing across episodes. In our preliminary experiment, this improved generalisation performance by making it difficult for LPG to exploit environment-specific patterns. Rectified linear unit (ReLU) was used as activation function throughout the experiment.

**Details of LPG Update**  In Section 3.3, the meta-gradient for updating LPG is described as the outcome of REINFORCE for simplicity. In practice, however, we used advantage actor-critic (A2C) [6] to calculate the meta-gradient, which requires learning value functions for bootstrapping. Note that value functions were trained only to reduce the variance of meta-gradient. LPG itself has no access to value functions during meta-training and meta-testing. In principle, the *outer* algorithm used for discovery can be any RL algorithm, as long as they are designed to maximise cumulative rewards.

**Details of Hyperparameter Balancing**  As described in Section 3.4, we trained a bandit $p(\alpha|\mathcal{E})$ to automatically sample better agent hyperparameters for each environment to make meta-training more stable. More specifically, the bandit samples hyperparameters at the beginning of each lifetime according to:

$$p(\alpha|\mathcal{E}) \propto \exp\left(\frac{R(\alpha, \mathcal{E}) + \rho/\sqrt{N(\alpha, \mathcal{E})}}{\tau}\right), \tag{1}$$

where $R(\alpha, \mathcal{E})$ is the final return at the end of the agent's lifetime with hyperparameters $\alpha$ in environment $\mathcal{E}$, which is averaged over the last 10 lifetimes. $N(\alpha, \mathcal{E})$ is the number of lifetimes simulated. $\tau$ is a constant temperature, and $\rho$ is a coefficient for exploration bonus. Intuitively, we keep track of how well each $\alpha$ performs and sample hyperparmeters that tend to produce a larger final return with exploration bonus. In our experiments, $\alpha$ consists of two hyperparameters: learning rate ($\alpha_{\text{lr}}$) and KL cost ($\alpha_y$) for updating the agent's predictions. Table 2 shows the range of hyperparameters searched by the bandit. Note that this hyperparameter balancing requires multiple lifetimes of experience, which can be done only during meta-training. During meta-testing on unseen environments, $\alpha$ needs to be manually selected.

**Preventing Early Divergence**  We found that meta-training can be unstable especially early in training, because the randomly initialised update rule ($\eta$) tends to make agents diverge or deterministic, which eventually causes exploding meta-gradients. To address this issue, we reset the lifetime whenever the entropy of the policy becomes 0, which means the policy becomes deterministic. We observed that this is triggered a few times early in training but eventually is not triggered later in training as the update rule improves.

Table 1: Meta-hyperparameters for meta-training.

| Hyperparameter | Value | Searched values |
|---|---|---|
| Optimiser | Adam | - |
| Learning rate | 0.0001 | $\{0.0005, 0.0001, 0.00003\}$ |
| Discount factor ($\gamma$) | $\{0.995, 0.99\}$ | - |
| Policy entropy cost ($\beta_0$) | $\{0.01, 0.02\}$ | - |
| Prediction entropy cost ($\beta_1$) | 0.001 | $\{0.001, 0.0001\}$ |
| L2 regularisation weight for $\hat{\pi}$ ($\beta_2$) | 0.001 | $\{0.01, 0.001\}$ |
| L2 regularisation weight for $\hat{y}$ ($\beta_3$) | 0.001 | $\{0.01, 0.001\}$ |
| Bandit temperature ($\tau$) | 0.1 | $\{1, 0.1\}$ |
| Bandit exploration bonus ($\rho$) | 0.2 | $\{1, 0.2\}$ |
| Number of steps for each trajectory | 20 | - |
| Number of parameter updates ($K$) | 5 | - |
| Number of parallel lifetimes | 960 | - |
| Number of parallel environments per lifetime | 64 | - |

Discount factor and policy entropy cost are randomly sampled from the specified range for each lifetime.

Table 2: Agent hyperparameters for each training environment.

| Environment | Architecture | Optimiser | Learning rate ($\alpha_{\text{lr}}$) | KL cost ($\alpha_y$) | Lifetime |
|---|---|---|---|---|---|
| dense | Tabular | SGD | $\{20, 40, 80\}$ | $\{0.1, 0.5, 1\}$ | 3M |
| sparse | Tabular | SGD | $\{20, 40, 80\}$ | $\{0.1, 0.5, 1\}$ | 3M |
| long_horizon | Tabular | SGD | $\{20, 40, 80\}$ | $\{0.1, 0.5, 1\}$ | 3M |
| longer_horizon | Tabular | SGD | $\{20, 40, 80\}$ | $\{0.1, 0.5, 1\}$ | 3M |
| long_dense | Tabular | SGD | $\{20, 40, 80\}$ | $\{0.1, 0.5, 1\}$ | 3M |
| dense | C(16)-D(32) | Adam | $\{0.0005, 0.001, 0.002, 0.005\}$ | $\{0.1, 0.5, 1\}$ | 30M |
| long_horizon | C(16)-D(32) | Adam | $\{0.0005, 0.001, 0.002, 0.005\}$ | $\{0.1, 0.5, 1\}$ | 30M |
| small | D(32) | Adam | $\{0.0005, 0.001, 0.002, 0.005\}$ | $\{0.1, 0.5, 1\}$ | 30M |
| sparse | D(32) | Adam | $\{0.0005, 0.001, 0.002, 0.005\}$ | $\{0.1, 0.5, 1\}$ | 30M |
| very_dense | C(32-16-16)-D(256) | Adam | $\{0.0005, 0.001, 0.002, 0.005\}$ | $\{0.1, 0.5, 1\}$ | 30M |
| short | Tabular | SGD | $\{20, 40, 80\}$ | $\{0.1, 0.5, 1\}$ | 1M |
| short_noisy | Tabular | SGD | $\{20, 40, 80\}$ | $\{0.1, 0.5, 1\}$ | 1M |
| long | Tabular | SGD | $\{20, 40, 80\}$ | $\{0.1, 0.5, 1\}$ | 1M |
| long_noisy | Tabular | SGD | $\{20, 40, 80\}$ | $\{0.1, 0.5, 1\}$ | 1M |
| distractor | D(16) | Adam | $\{0.002, 0.005, 0.01\}$ | $\{0.1, 0.5, 1\}$ | 2M |

'C(N1-N2-...)' represents convolutional layers with N1, N2, ... filters for each layer.
'D(N)' represents a dense layer with N units.
Lifetime is defined as the total number of steps.

## B.2 Meta-Testing

We selected the best update rule ($\eta$) and hyperparameters according to the validation performance on two Atari games (`breakout, boxing`), and used them to evaluate across all 57 Atari games. We found that subtracting a baseline slightly improves the performance on Atari games as follows:

$$\Delta\theta \propto \mathbb{E}_{\pi_\theta}\left[\nabla_\theta \log \pi_\theta(a|s)(\hat{\pi} - f_\theta(s)) - \alpha_y\nabla_\theta D_{\text{KL}}(y_\theta(s)\|\hat{y}) - \frac{1}{2}\|f_\theta(s) - \hat{\pi}\|^2\right], \quad (2)$$

where $f_\theta(s)$ is an action-independent baseline function. The hyperparameters are summarised in Table 3, and the learning curves are shown in Figure 1.

## B.3 Computing Infrastructure

Our implementation is based on JAX [1], RLAX [2], Optax [4], Haiku [3] using TPUs [5]. The training environments are also implemented in JAX, which enables running on TPU as well. It took approximately 24 hours to converge using a 16-core TPU-v2.

Table 3: Hyperparameters used for meta-testing on Atari games.

| Hyperparameter | Value | Searched values |
|---:|---|---|
| Optimiser | Adam | - |
| Network architecture | C(32)-C(64)-C(64)-D(512) | - |
| Learning rate ($\alpha_{\text{lr}}$) | 0.0005 | {0.001, 0.0005, 0.0003} |
| KL cost ($\alpha_{\text{y}}$) | 0.5 | {1, 0.5, 0.1} |
| Discount factor ($\gamma$) | 0.995 | - |
| Number of steps for each trajectory | 20 | - |
| Number of parallel environments (batch size) | 30 | - |

# C   Generalisation to Atari Games

Figure 1: Learning curves on Atari games. X-axis and y-axis represent the number of frames and episode return respectively. Shaded areas show standard errors from 3 independent runs.