[Reviews · NeurIPS 2020]

Review 1

Summary and Contributions: This is a paper where they basically try to discover new RL algorithms. The approach is simple: they use a gradient-based direct policy search technique to solve a Bayesian type of RL problem with a rather rich hypothesis space where every element corresponds to a "classical vector of parameters" plus what we could call a RL algorithm.

Strengths: They address a very relevant problem, namely the automatic discovery of RL algorithms. But such an approach for learning RL algorithms has already been used before. The main contribution of this paper would be the way they define the candidate space for the RL algorithms, but this is never well explained in the paper.

Weaknesses: They did not do a good review of the related work. Problem badly formalized. Main contribution - the way they define the set of RL algorithms - not put forward in a proper way. Simulation results carry out on fairly simple problems and not that convincing.

Correctness: Yes

Clarity: Not that well because they do not position clearly their contribution.

Relation to Prior Work: Not well done. See section additional feedback and comments.

Reproducibility: Yes

Additional Feedback: Page 2: In your related work, you have missed several important works, such as for example those of Francis Maes where he proposes approaches for learning fundamental learning rules for RL algorithms (especially for playing bandit problems), see https://scholar.google.be/citations?hl=fr&user=h8kelPwAAAAJ His approach is very close to yours (same type of objective function). Page 3: The finding of an optimal update policy is in some sense expressed as a Bayesian RL problem (you know a probability distribution over environments as prior) but you never make the connection with this field of research. That’s a pitty. In the work of Maes, it is somehow formalized as such. Page 4. You approach can be considered as a gradient-based direct policy search approach for which you have as evaluation metric formula (1), as search space \eta \times \theta and as optimization method a gradient-based method. The main contribution of this paper is how to define the candidate space of your \neta, something you never define very well. That’s a pitty.


Review 2

Summary and Contributions: **Update : No new experiments with continuous control have been added, and so I am not increasing my score. Proposes an algorithm that learns an entire update rule for RL (without relying on value functions like prior work). The update rule is parameterized via an LSTM, which produces updates for the policy weights and a semantic vector (which is shown to converge to a notion of the value function), for N timesteps in a lifetime. Meta-learning is done by considering the performance of a population of agents each with its own policy weights, semantic vector and in separate environments, but sharing the same update rule. Authors show that the RL procedure obtained after meta-training on a set of grid worlds can beat human level performance on 14 atari games.

Strengths: 1) The presented approach is able to learn an entire RL update rule. The only prior meta-RL work that can learn an RL rule effective for an entirely different environment is MetaGenRL [1], but it still uses TD learning for value functions (only the policy update is learned). The authors of this work empirically show that their completely learned update gives better results. Further, there is a thorough analysis of the predicted semantic vector which shows that it does seem to correspond to the notion of a value function and the learned algorithm does seem to be implementing a form of bootstrapping. 2) Impressive empirical result of learning an RL rule from grid worlds that is effective on atari games. The plot of performance against the types of environments meta-trained seems to indicate that such methods can keep improving the learned RL algorithm with more data, making this a promising direction for future work. [1] : Improving Generalization in Meta Reinforcement Learning using Learned Objectives (Kirsh et al.)

Weaknesses: 1) The formulation isn't very novel in the sense that it reuses many aspects first introduced in prior work : it combines the popolution setup of MetaGenRL [1] for learning a shared update rule, with the LSTM parameterization of the update rule [2]. (it does however enforce a separation between the agent and the algorithm unlike prior works that used LSTMs). 2) While the existing evaluation is quite thorough on the chosen domains, it would be interesting to see how the learned rule performs on continuous control mujoco tasks/other robotic domains commonly tested in meta-RL, but this is not included. This also requires dealing with continuous state and action spaces, which would be good to address. [1] : Improving Generalization in Meta Reinforcement Learning using Learned Objectives (Kirsh et al.) [2] : RL2: Fast Reinforcement Learning via Slow Reinforcement Learning (Duan et al.)

Correctness: The claims and the method seem correct.

Clarity: The paper is well-motivated, clear and easy to follow.

Relation to Prior Work: The paper goes over previous meta-RL work quite well, categorizing methods based on how well the learned algorithm can generalize.

Reproducibility: Yes

Additional Feedback:


Review 3

Summary and Contributions: The authors introduce an approach for learning RL algorithms in which both the policy and prediction (analogous to the value function) are both updated by a meta-learned network. After training their approach on simple grid world environments the authors show the algorithm can generalize to Atari, where it achieves a competitive performance with A2C.

Strengths: The concept of learning both the policy update and prediction is interesting and a natural progression from previous results. The empirical results are particularly convincing, generalizing from the toy environments to a much more challenging domain, Atari. Furthermore, the approach appears to be surprisingly straightforward/elegant if one ignores all the (presumably) necessary minor details. The analysis and ablation on the method is informative and interesting and the supplementary material contains enough details to reproduce the method even without the code.

Weaknesses: The main distinction of this approach over previous results is the prediction/value function is learned as well as the policy update. The input to LPG is a 5-dim vector (r, terminal, gamma, y(s_t), y(s_t+1)) where the predictions y are reduced to a scalar. My main concern is that the information provided to LPG implicitly forces the prediction to be roughly the value function. Or, in other words, its unclear that LPG might predict anything very different than a function of future rewards. This makes me feel like the main result of the paper is closer to re-discovering RL algorithms rather than discovering new approaches, and the claim that the RL algorithm learns its own prediction is over-stated. The results with LPG-V possibly suggest otherwise but there was a lot of design choices made (regularizers, architecture, hyper-parameter tuning, etc) and unless LPG-V was as carefully optimized, it's not surprising that arbitrarily plugging in a VF learning rule would perform worse.

Correctness: The empirical methodology appears to be correct and useful analysis is provided. Each aspect of the approach seems well-motivated or studied with ablation.

Clarity: The paper is well-written, the approach is clear, and presentation quality is high (figures and paper organization).

Relation to Prior Work: There is a clear distinction from previous results and the approach feels like part of a clear progression in this research area.

Reproducibility: Yes

Additional Feedback: An interesting analysis would be to show the prediction output of LPG is sufficiently different from the Bellman equation. Additional Questions: Is passing gamma to the LPG necessary since the meta-objective G includes it already? Does LPG still learn well if the input to the LPG is augmented with additional information, such as a condensed representation of the state?

[Author Response · NeurIPS 2020]

We thank the reviewers for constructive feedback. We are glad that they found our idea of discovering an entire RL update rule interesting [**R2**, **R3**] and recognised our work as a clear distinction from the previous work [**R3**] and a promising direction [**R2**]. We are also encouraged that they found our empirical results impressive [**R3**] and convincing [**R2**]. We address questions raised by each reviewer below and will incorporate some of the feedback in the revision.

[**R1**] **"But such an approach for learning RL algorithms has already been used before." "They did not do a good review of the related work." "missed several important works, such as for example those of Francis Maes ..."**
Though we thank the reviewer for pointing out some relevant work, we would like to clarify the difference and potential misunderstandings. First of all, our approach is new and distinct from the prior work in that there has been no prior approach that attempts to discover alternatives to value functions and TD-learning as agreed by [**R2**, **R3**]. Secondly, Maes et al. [1], [2] aim to find the formula of the Bayes-optimal policy given hand-designed variables (e.g., $Q(s,a), N(s,a)$) by searching over math operations $\mathcal{F}$, where $p(a|s) = \mathcal{F}(\{Q(s,a), N(s,a)\})$. Thus, the update rules for the variables are **NOT** discovered but hand-designed in Maes et al., whereas our approach discovers an update rule for them: what these variables should be and how to update them. Finally, although our problem could be interpreted as a Bayesian inference problem, it is quite different from the conventional Bayesian RL, because the update rule does not directly interact with the environment unlike the policy in Maes et al. We will cite and discuss them in the revision.

[**R1**] **"Problem badly formalized." "The main contribution of this paper is how to define the candidate space of your $\eta$, something you never define very well."**
Our problem and its objective is clearly defined in Eq. 1. In addition, unlike Maes et al. [1], [2], where the solution space can be expressed by a finite set of operators, the update rule is parameterised by a neural network $\eta$ in our work. Thus, the solution space of $\eta$ is determined by the network architecture, which is also clearly defined in the paper.

[**R1**] **"Simulation results carry out on fairly simple problems and not that convincing."**
ALE (Atari games) is a very challenging benchmark. We believe that the fact that the update rule discovered solely from toy domains can generalise well to Atari games is very interesting and convincing as agreed by [**R2**, **R3**].

[**R1**] **"Main contribution - the way they define the set of RL algorithms - not put forward in a proper way."**
We respectfully disagree. As emphasised in the paper and acknowledged by [**R2**, **R3**], our main contribution is rather to show that it is possible to discover an entire update rule that can replace value functions and TD-learning, and that the update rule discovered from toy domains can generalise surprisingly well to complex Atari games.

[**R2**] **Regarding novelty of the formulation**
We would like to re-emphasise that our formulation is novel in that it is the first to discover alternatives to value functions. In terms of meta-training, we would like to point out that our method is not just a combination of the prior work. Since the problem of discovering an entire update rule is much more challenging than discovering only a policy update rule as in MetaGenRL, we had to develop several new methods (e.g., regularisers, balancing hyperparameters), which turned out to be crucial (see ablation study). We believe that they would be useful for the future work as well.

[**R3**] **Regarding whether LPG predicts something beyond future rewards and LPG-V**
We agree that predictions are implicitly encouraged to be a function of future rewards. However, we claim that there is still significant room for improvement by discovering a better form of such a function. For example, variations of TD-learning methods (e.g., distributional RL, $\Gamma$-net, mixtures of $n$-step and $\lambda$-returns, non-linear reward transformation) have been shown to perform quite differently, even though the underlying principle is the same. We suspect that LPG could discover a more efficient class of TD-learning and even beyond. In fact, Figure 5 shows that LPG captures values at various discount factors, which is already interesting in that none of the RL algorithms maintains values at various discount factors at the same time. This indicates that LPG is doing something different from TD-learning for more efficient bootstrapping. To further clarify, $y$-vector is mapped to a scalar only within the update rule, but $y$ still maintains richer information compared to the value function, which is why it can capture values at various horizons. Finally, we picked the best LPG-V by tuning it with and without all the tricks (regularisers, balancing hyperparameters). Thus, we believe that the result shows that the discovered prediction semantics is crucial for the performance. In fact, this is also consistent with MetaGenRL's result, where the discovered policy update rule (given value functions) does not outperform the baseline DDPG even on the training environments.

[**R3**] **Is passing $\gamma$ necessary? Does LPG still learn with additional information such as state representation?**
Yes. $\gamma$ is treated as a part of the environment, which is varied during training. So, the optimal update rule depends on the given $\gamma$. We expect that giving domain-specific information like state representation is likely to improve LPG on training domains but hurt on unseen domains, which is why we intentionally removed such information. However, adding such information without compromising generalisation performance would be a very interesting future direction.

[1]  F. Maes, L. Wehenkel, and D. Ernst, "Automatic discovery of ranking formulas for playing with multi-armed bandits," in *European Workshop on Reinforcement Learning*, Springer, 2011, pp. 5–17.

[2]  M. Castronovo, F. Maes, R. Fonteneau, and D. Ernst, "Learning exploration/exploitation strategies for single trajectory reinforcement learning," in *European Workshop on Reinforcement Learning*, 2013, pp. 1–10.


[Meta-Review · NeurIPS 2020]

2/3 reviewers believe the paper will be important to the meta reinforcement learning community, and recommend acceptance. The third recommended rejection, but did not argue for rejection in the discussion. Despite the overall positive response, the reviewers shared R1's concerns about missing related work.